# Continuous synthesis of high-entropy alloy nanoparticles by in-flight alloying of elemental metals

Keun Su Kim [1,2,3] ✉, Martin Couillard[4], Ziqi Tang [3], Homin Shin[1], Daniel Poitras [5], Changjun Cheng [6], Olga Naboka[7], Dean Ruth[1], Mark Plunkett[1], Lixin Chen[6], Liliana Gaburici[1], Thomas Lacelle [1], Michel Nganbe [3] & Yu Zou [6]

High-entropy alloy (HEA) nanoparticles (NPs) exhibit unusual combinations of functional properties. However, their scalable synthesis remains a significant challenge requiring extreme fabrication conditions. Metal salts are often employed as precursors because of their low decomposition temperatures, yet contain potential impurities. Here, we propose an ultrafast (< 100 ms), one-step method that enables the continuous synthesis of HEA NPs directly from elemental metal powders via in-flight alloying. A high-temperature plasma jet (> 5000 K) is employed for rapid heating/cooling ($10^3 - 10^5$ K s$^{-1}$), and demonstrates the synthesis of CrFeCoNiMo HEA NPs (~ 50 nm) at a high rate approaching 35 g h$^{-1}$ with a conversion efficiency of 42%. Our thermofluid simulation reveals that the properties of HEA NPs can be tailored by the plasma gas which affects the thermal history of NPs. The HEA NPs demonstrate an excellent light absorption of > 96% over a wide spectrum, representing great potential for photothermal conversion of solar energy at large scales. Our work shows that the thermal plasma process developed could provide a promising route towards industrial scale production of HEA NPs.

High-entropy alloys (HEAs), consisting of five or more principal elements with a near equimolar ratio, have become one of the most transformative concepts in current alloy design[1–3]. Homogenous mixing of a large number of elements induces synergistic effects among different elements, thus resulting in an unusual combination of functional properties appealing to a broad range of applications such as structural alloys, catalysis, sensing, and energy storage[4]. While the synthesis of bulk HEAs has been the main focus in the past decade, nano-sized HEA particles are emerging as a new class of multifunctional materials due to their more fascinating properties[5]. Scalable and economically viable synthetic methods for HEA nanoparticles

(NPs) are of particular interest, yet the controlled incorporation of multiple elements into a tiny particle (<100 nm) remains a significant challenge[5].

In 2018, a carbothermal shock (CTS) technique was developed to incorporate multiple immiscible metal elements into a single NP and successfully demonstrated the synthesis of HEA NPs containing up to eight elements uniformly dispersed on a conductive carbon support[6]. Although the CTS technique has demonstrated a remarkable potential in synthesizing HEA NPs in a controlled manner, the process is limited to electrically conductive supports and operated in a batch mode, thus not suitable for industrial-scale production of HEA NPs. To overcome

[1]Security and Disruptive Technologies Research Centre, National Research Council Canada, Ottawa, ON K1A 0R6, Canada. [2]Department of Mechanical and Industrial Engineering, University of Toronto, Toronto, ON M5S 3G8, Canada. [3]Department of Mechanical Engineering, University of Ottawa, Ottawa, ON K1N 6N5, Canada. [4]Energy, Mining and Environment Research Centre, National Research Council Canada, Ottawa, ON K1A 0R6, Canada. [5]Advanced Electronics and Photonics Research Centre, National Research Council Canada, Ottawa, ON K1A 0R6, Canada. [6]Department of Materials Science and Engineering, University of Toronto, Toronto, ON M5S 3G8, Canada. [7]Construction Research Centre, National Research Council Canada, Ottawa, ON K1A 0R6, Canada. ✉e-mail: KeunSu.Kim@nrc-cnrc.gc.ca

this challenge, new methods building on the CTS technique were reported, such as microwave heating[7], fast moving bed pyrolysis[8], and aerosol methods[9]. Very recently, a pulsed/scanning laser ablation method was also reported and demonstrated synthesis of high-entropy materials on various substrates[10]. However, the feedstock is typically limited to metal salts. The vapor-source technique also provides a very effective way to form alloy NPs. In this approach, a well-mixed vapor comprising multiple elements is formed from the vaporization of pure metal feedstock, and then rapidly quenched to form crystal solids. High-temperature (>4000 K) environments are usually created by using arc discharge[11] and oscillatory spark discharge[12]. Similar to the CTS, this approach has demonstrated great potential for rapid synthesis of various HEA NPs. However, most processes are operated in a batch mode. The current state-of-the-art technology is still lacking a scalable synthesis method that enables the continuous synthesis of HEA NPs directly from a mixture of pure elemental metal feedstock.

To address the above challenge, we propose an ultrafast (<100 ms), one-step method for the continuous synthesis of HEA NPs directly from a mixture of pure elemental metal powders by using a thermal plasma jet. Thermal plasma jets are partially ionized gases with high temperature (>8000 K) and high speed[13]. They are capable of rapid heating of feedstock to produce an atomically mixed multi-component vapor, and also rapid cooling of the vapor at an ultrahigh cooling rate of $10^5 - 10^6$ K s$^{-1}$ to form solid-solution particles. Feedstock can be injected continuously, thus this technology is well suited for scalable synthesis of HEA NPs[14]. Despite this unique potential, thermal plasma jets have only been used for the processing of bulk HEAs[15] or spheroidization of pre-alloyed HEA powders[16]. The main challenge would be the existence of a large nucleation temperature gap among different metal vapors; upon plasma jet cooling, the element with the lowest saturation vapor pressure reaches the supersaturated state first and segregates.

Here, in a proof-of-concept demonstration, we report the synthesis of CrFeCoNiMo HEA NPs with an average size of 50 nm at a high rate approaching 35 g h$^{-1}$ using an inductively coupled plasma jet (ICPJ). Despite the low saturation pressure of Mo, all metal elements in a particle are homogeneously mixed at the atomic level with good crystallinity. It is discussed that mixing multiple elements at a near equimolar ratio not only increases the mixing entropy but also decreases the partial pressure of the constituent element (i.e.,

$P_i = P_{total}/N$, where $P_i$ is the partial pressure of $i$th constituent element, $P_{total}$ is the total pressure, and $N$ is the total number of the constituent elements) in the vapor, which prohibits the continuous growth of nuclei into pure metal particles via homogenous condensation. This renders the HEA NPs formation feasible even in the presence of a large nucleation temperature gap. We have also investigated the effects of reactor geometry and plasma gases on the NP growth using homogenous nucleation theory and thermofluid simulation. This synthetic route presents considerable potential in the commercial scale applications of HEA NPs where a large amount of powders are desirable. The as-synthesized HEA NPs are exploited as high-performance photo-thermal materials for solar energy harvest, and achieve a high absorptance of >96% over a wide spectrum without noble metals.

## Results

### Thermal plasma synthesis of HEA NPs

Figure 1a shows the schematics of the plasma process developed for the continuous synthesis of HEA NPs from a mixture of pure elemental metal powders (see also Supplementary Fig. 1). In a typical synthesis run, a mixture of five elemental metal powders at an equimolar ratio is continuously fed into a high-temperature plasma jet generated by an induction plasma torch (45 kW, ~3 MHz, 66.7 kPa). The feed rate typically ranges from 1.2 g min$^{-1}$ to 2.0 g min$^{-1}$. The injected feedstock evaporates within a few tens of milliseconds in the core of the plasma jet (>8000 K) releasing elemental metal vapors. The metal vapors are mixed together upon plasma jet expansion and form a homogeneous multicomponent vapor in the reactor zone. As a rapid quenching is applied, nuclei are formed from the multicomponent vapor and particles grow through the co-condensation of metal monomers. The final HEA NPs formed are continuously collected by a porous filter unit at a high yield rate of ~35 g h$^{-1}$, while unvaporized feedstock is removed from the reaction stream by a cyclone separator.

To ensure the compositional uniformity, unlike the conventional alloy (Fig. 1b), it would be ideal that all the constituent elements in the vapor nucleates simultaneously by co-nucleation process (i.e., nucleation by multicomponent), followed by co-condensation of metal monomers onto the nuclei formed (Fig. 1c). However, the nucleation temperature of a metal vapor strongly depends on its saturation vapor pressure - an intrinsic property of an element. This presents a potential challenge in the synthesis of HEA NPs from the direct vaporization of metal powders by a plasma jet. As a proof-of-concept demonstration,

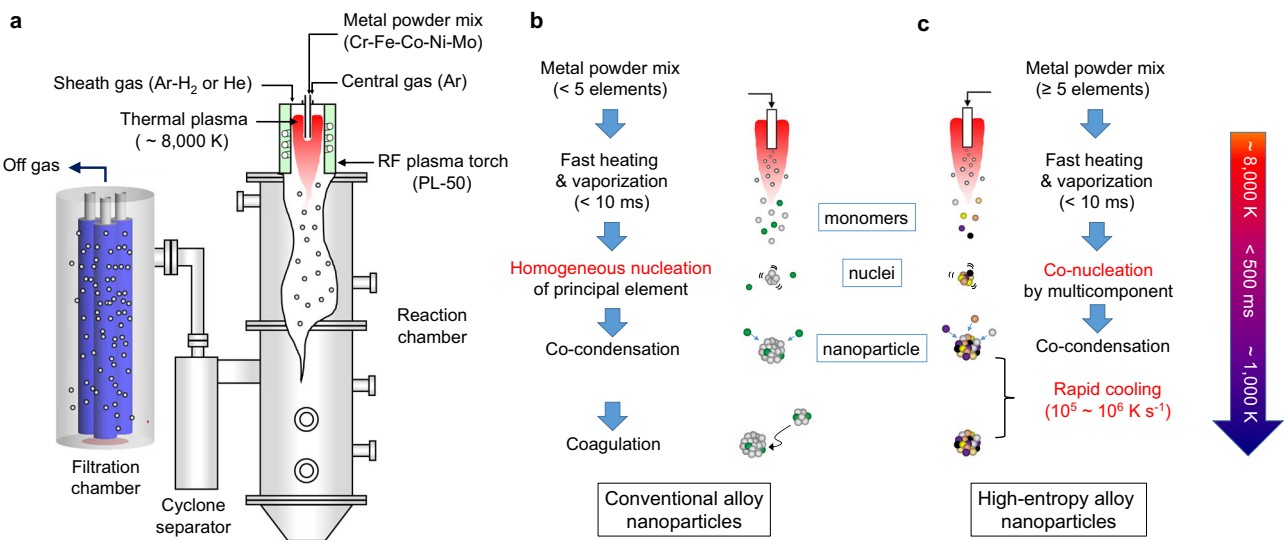

**Fig. 1 | The ICPJ strategy for continuous synthesis of HEA NPs. a** Schematic of an inductively coupled plasma jet (ICPJ) process developed for the continuous synthesis of HEA NPs directly from a mixture of pure elemental metal powders via in-flight alloying. **b, c** Schematic diagrams illustrating the formation mechanism of (**b**) conventional alloy NPs and (**c**) HEA NPs by a thermal plasma jet.

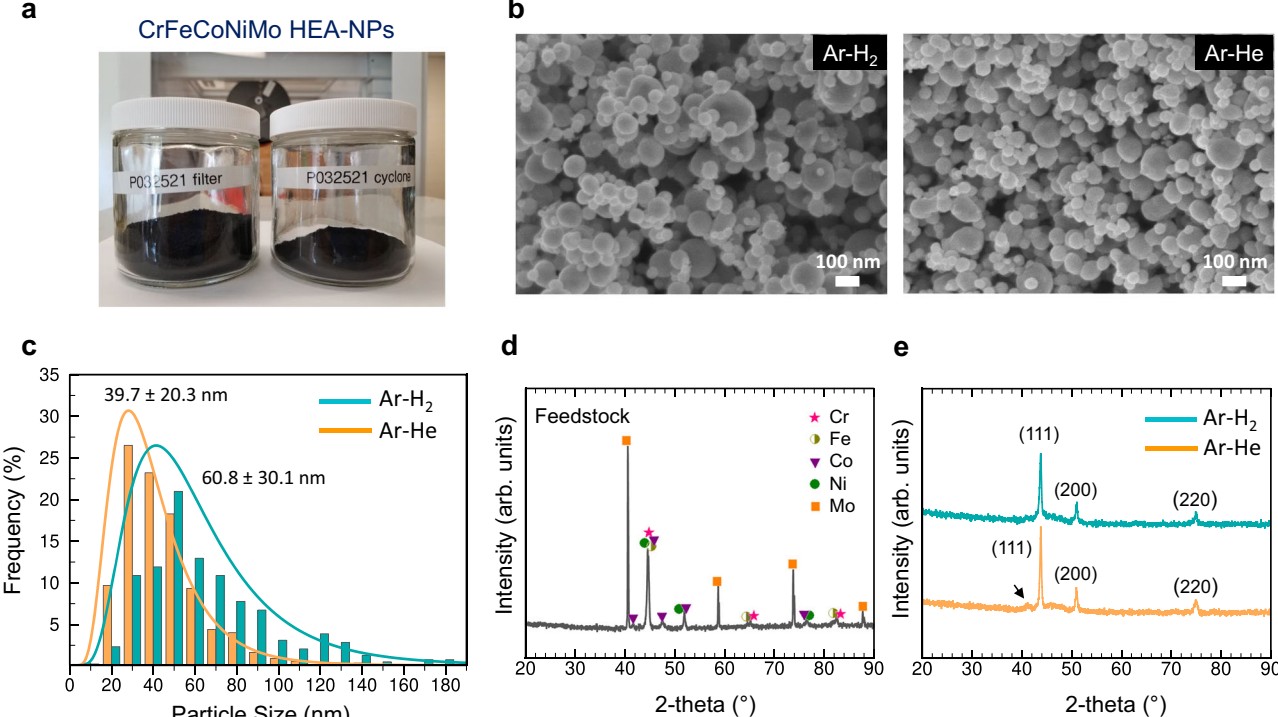

**Fig. 2 | Morphological and structural characterization of HEA NPs. a** Photo of the HEA NP samples produced after a 150-min synthesis experiment. **b** SEM images of the HEA NPs produced with different plasma gases of hydrogen and helium, showing morphology change. Scale bar, 100 nm. **c** Size distribution of the HEA NPs produced with different plasma gases. **d** XRD pattern of the feedstock mixture of Cr, Fe, Co, Ni, and Mo. **e** XRD patterns of the HEA NPs (single FCC) produced with different plasma gases which confirm the in-situ alloying of pure elemental metals by the ICPJ strategy.

we have studied a Cr-Fe-Co-Ni-Mo system where Mo exhibits a significantly different nucleation temperature compared with other elements, and discussed the growth mechanism. To investigate the effects of the heating/cooling rate, we have also employed plasma gases with high thermal conductivity such as hydrogen and helium. These NPs are denoted as HEA-H$_2$ and HEA-He, respectively in the following discussion. A plasma gas comprising 100% Ar is also of interest for the comparison; however, it was not considered in this work because pure argon gas is limited to operations at low plasma powers (<25 kW) to avoid damage to the plasma torch.

### Morphology, composition, and structure characterization

Figure 2a shows a photo of the HEA-H$_2$ sample collected from the cyclone separator (42 g) and the filter unit (84 g) after a 150-min operation. In total about 200 g of powder was fed and the productivity approaches 35 g h$^{-1}$ (with a conversion rate of 42%, Supplementary Table 1), which is >10 times higher than what has been reported in the literature to date[17]. The productivity or the conversion rate can be further improved by minimizing powder deposition on the reactor walls (74 g), such as by employing porous reactor walls that allow continuous gas flow through the wall surfaces. Figure 2b presents scanning electron microscopy (SEM) images of the as-synthesize powders collected from the filter unit, showing the effect of the plasma gases (Supplementary Fig. 2 shows more SEM images). In both cases, the samples consist of spherical NPs, dispersed uniformly without serious physical aggregation/agglomeration. However, the HEA-He sample exhibits a smaller average size with a narrow size distribution. The average diameter is estimated as 39.7 ± 20.3 nm for the HEA-He sample while 60.8 ± 30.1 nm for the HEA-H$_2$ sample from transmission electron microscopy (TEM) images (Supplementary Fig. 3). This may be attributed to the different cooling rate and residence time achieved by the different plasma gases.

The thermal plasma process is usually accompanied by the production of impurities from incomplete vaporization of feedstock because a fraction of the feedstock injected typically bounces back from the plasma core. This has also been prevalent in the ICPJ process due to the elevated viscosity near the plasma core (e.g., 10 times higher than at room temperature)[18] and the presence of recirculation eddies formed by the magnetic pinch[19]. Unprocessed feedstock powders are larger than NPs, and thus can be removed from the reaction stream by means of centrifugal force. SEM images of the sample collected from the cyclone separator confirmed the presence of large particulates up to a few tens of μm (Supplementary Fig. 2a). However, such particulates are not observable from the samples collected from the filter unit (Supplementary Fig. 2b, c), suggesting that the cyclone separator is highly effective in removing the unprocessed powder in the current ICPJ process.

The X-ray diffraction (XRD) pattern of the feedstock in Fig. 2d confirms the existence of the five elements, showing the peaks corresponding to their different crystal structures (Cr: BCC, Fe: BCC, Co: HCP, Ni: FCC, and Mo: BCC); however, the patterns of the reaction products exhibit a single FCC structure with diffraction peaks at 43.3° 50.4° and 74.1°, corresponding to the (111), (200) and (220) planes of a FCC structure, respectively (Fig. 2e). An additional small peak also showed up around 41°, but this is possibly attributable to the oxide formation at the surface. This result clearly supports that the mixture of metal powders injected was successfully alloyed in the ICPJ process. The sharp peak at 43.3° reveals the high crystallinity of the HEA samples produced. The $d$-spacing of each sample was estimated and summarized in Supplementary Tables 2, 3 with calculated lattice constants. While both samples exhibit a similar average lattice constant around 3.58 Å ($a_O = 3.63$ Å for Cr$_{0.2}$Fe$_{0.2}$Co$_{0.2}$Ni$_{0.2}$Mo$_{0.2}$ from the density functional theory (DFT) simulation, Supplementary Fig. 4), the HEA-He exhibits a larger crystallite size of $L_a = 17.93$ nm compared to

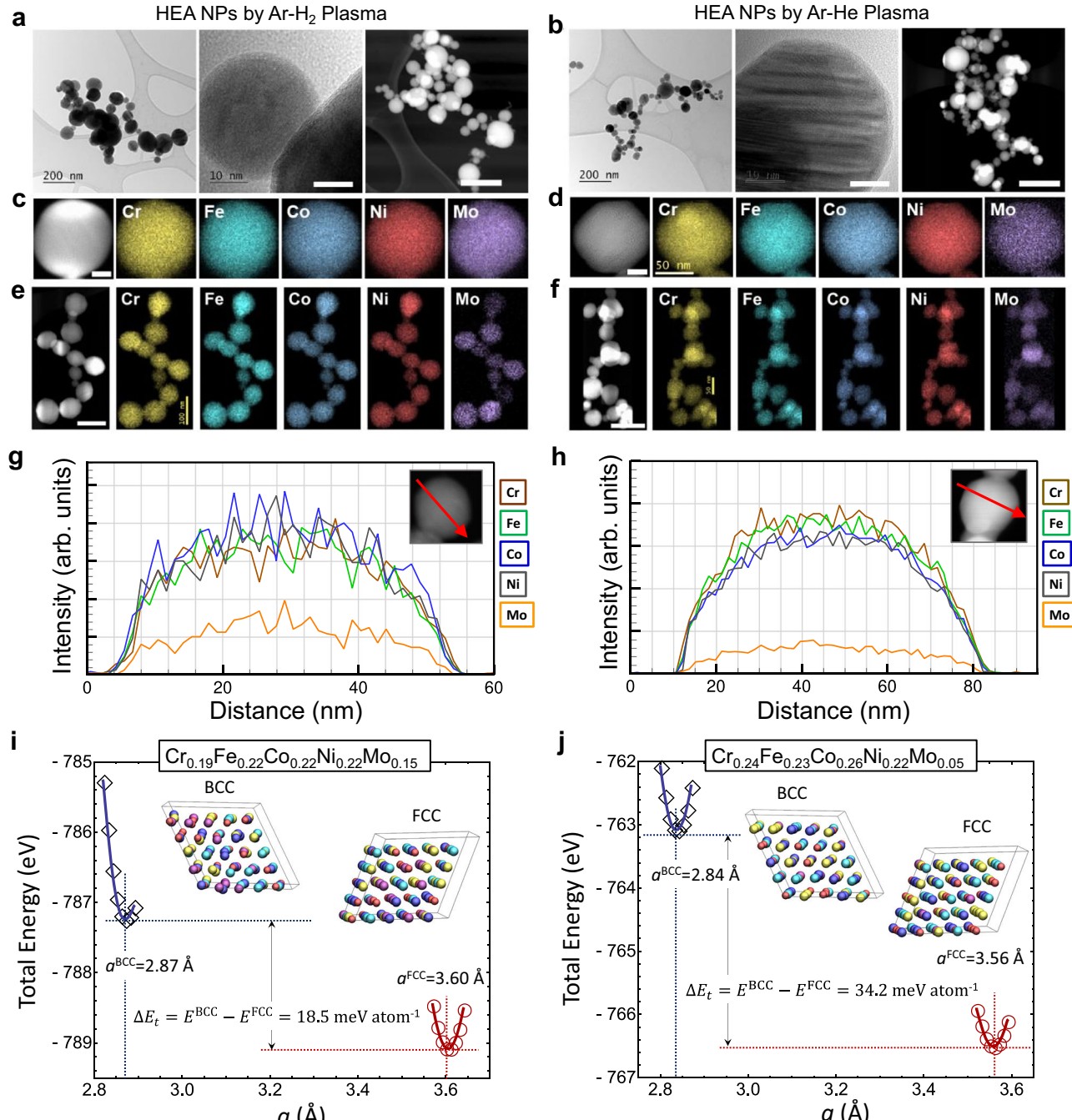

**Fig. 3 | Composition and phase stability analysis of HEA NPs. a, b** TEM, HR-TEM and HAADF-STEM images of the HEA NPs produced with different plasma gases of hydrogen and helium by the ICJP. Scale bar, 10 nm (HR-TEM) and 100 nm (HAADF). **c–f** EDX elemental maps of single and multiple HEA NPs, showing homogenous distribution of the five metals in particles. Scale bar, 25 nm (single NP) and 100 nm (multiple NPs). **g, h** EDX line scans of individual NPs showing the spatial uniformity in their compositions; (**g**) HEA-H$_2$ case and (**h**) HEA-He case. **i, j** Phase stability calculations by DFT simulation for the HEA NPs; (**i**) Cr$_{0.19}$Fe$_{0.22}$Co$_{0.22}$Ni$_{0.22}$Mo$_{0.15}$ (HEA-H$_2$ case) and (**j**) Cr$_{0.24}$Fe$_{0.23}$Co$_{0.26}$Ni$_{0.22}$Mo$_{0.05}$ (HEA-He case), demonstrating a higher stability of a FCC structure over a BCC.

HEA-H$_2$ ($L_a = 17.08$ nm), probably due to the different cooling rate employed. The XRD patterns of the samples collected from the cyclone are shown in Supplementary Fig. 5. Co-existence of HEA NPs and feedstock material is evident from the patterns, which is consistent with the SEM observation. Among various metal peaks, the Mo peak still remains comparable to those of HEA, indicating Mo might have been evaporated less compared with other elements. This may present a challenge to reuse those powders because the ratio among different elements has changed. The vaporization efficiency should be improved through optimization of the processing parameters (e.g.,

feed rate, plasma power, powder injection geometry and so on) so that the amount of powders collected in the cyclone can be minimized.

Figure 3 presents representative high-resolution TEM (HR-TEM) and high-angle annular dark-field scanning TEM (HAADF-STEM) images of the samples with energy dispersive X-ray (EDX) spectroscopy elemental mapping and EDX line scans of individual particles. Regardless of the plasma gases employed, homogenous mixing of five elements in a single particle (~60 nm) was observed without significant elemental segregation or phase separation; however, a slight segregation of Cr is visible from the HEA-He sample (see also Supplementary

Fig. 6), implying that the two samples were grown under different environments or via different growth mechanisms. The presence of oxygen in the HEA NP was also evident from the elemental mapping (Supplementary Fig. 7); however, the presence of oxygen is limited to the surface of the NP (~5 nm thick). The oxide layers might have been formed upon exposure to air. EDX line scans of individual particles for each HEA sample are presented in Fig. 3g, h, which confirm the spatial uniformity in their compositions throughout the particle. A statistical study has been performed to evaluate the composition of the HEA NPs produced (Supplementary Figs. 8–10 and Supplementary Table 4). The average composition was estimated as Cr (19.2%), Fe (21.5%), Co (22.3%), Ni (22.1%), and Mo (14.7%) for HEA-$H_2$, while Cr (23.9%), Fe (23.3%), Co (25.6%), Ni (22.0%), and Mo (5.2%) for HEA-He, which differ from that of the feedstock, especially for its Mo composition. This is likely due to the relatively low evaporation pressure of Mo (e.g., 0.0107 kPa at 3000 K) compared to those of the other four elements (e.g., Cr: 132 kPa, Fe: 67.3 kPa, Co: 40.4 kPa, and Ni: 52.8 kPa at 3000 K)[20]. Molybdenum might have evaporated less in the plasma jet, especially in the helium plasma case. A similar result has been reported on the HEA NPs synthesis (e.g., Co-Cr-Cu-Fe-Ni system) with the arc discharge technique[11], where elements with higher evaporation pressures (e.g., Cu) become rich in the products. To achieve the targeted composition of a near equimolar ratio, we could increase the plasma power to ensure the complete vaporization of Mo in feedstock or increase the Mo content in feedstock to compensate its low evaporation pressure[7]. The composition also varied among NPs with a variation up to about 8.4% for HEA-$H_2$ and 2.4% for HEA-He, respectively. This may be attributable to the relatively large particle size distribution. Further process optimization in this regard is needed to achieve good uniformity.

Crystal structure of the HEA NPs was also investigated by atomically-resolved HAADF-STEM image and the corresponding Fast Fourier Transform (FFT) analysis (Supplementary Fig. 11). Although most of the constituent elements have a BCC structure (Supplementary Table 5), the analysis reveals that the HEA NPs were stabilized with a FCC structure. DFT calculations were performed to investigate the phase structures for the two different compositions that were identified by HR-TEM analysis: $Cr_{0.19}Co_{0.22}Fe_{0.22}Ni_{0.22}Mo_{0.15}$ (HEA-$H_2$) and $Cr_{0.24}Co_{0.23}Fe_{0.26}Ni_{0.22}Mo_{0.05}$ (HEA-He). As shown in Fig. 3i, j, the relative stability of the BCC and FCC phases were estimated by evaluating the total energy differences between the BCC and FCC phases, $\Delta E_t = E^{BCC} - E^{FCC}$, and the equilibrium lattice constants were obtained from the calculation of the total energy as a function of lattice constant $a$ for each phase. For both compositions, $\Delta E_t$ was found to be positive, indicating that the FCC structure is more stable. Atomic lattice distortion appeared more notable for the BCC phase as presented in the relaxed structures in Fig. 3i, j. This observation was also quantitatively confirmed by the lattice distortion energy $\Delta E_{dist}$ (defined as the difference between the energies calculated with and without relaxation) estimated as $\Delta E_{dist}^{BCC} = 60.7$ and $\Delta E_{dist}^{FCC} = 37.9$ meV atom$^{-1}$ for HEA-$H_2$ and $\Delta E_{dist}^{BCC} = 30.9$ and $\Delta E_{dist}^{FCC} = 21.0$ meV atom$^{-1}$ for HEA-He. It was found that as the atomic fraction of Mo decreases from 15 to 5%, the FCC phase stability increases from $\Delta E_t = 18.5$ to 34.2 meV atom$^{-1}$, while the equilibrium lattice constant decreases from $a = 3.60$ to 3.56 Å. Note that the VEC value (VEC = $\sum c_i VEC_i$, where $c_i$ is the atomic fraction of the alloy component $i$) increases from 7.98 to 8.12, in agreement with the DFT calculation indicating that the FCC phase becomes more stable for the smaller Mo content. The FCC formation enthalpy, defined as the difference between the total energy of HEA and the ground-state energy of the $i$ component $E_i$, $\Delta H = E_{HEA} - \sum c_i E_i$, were calculated as 114.6 and 86.4 meV atom$^{-1}$ for the HEA-$H_2$ and HEA-He, respectively. The BCC formation enthalpy was as estimated 133.1 and 120.6 meV atom$^{-1}$ for the HEA-$H_2$ and HEA-He, respectively. The mixing entropy ($\Delta S_{mix} = -k_B \sum_i c_i \ln c_i$) contributions for the HEA-$H_2$ and HEA-He compositions are $1.60 k_B$ and $1.51 k_B$, where $k_B$ is the Boltzmann

constant. The temperature that gives the HEA FCC phase formation (i.e., the negative Gibbs free energy $G = \Delta H - T\Delta S_{mix}$) is 832 K and 662 K for HEA-$H_2$ and HEA-He, respectively. The calculation implies that though the HEA formation is driven by mixing entropy as the formation enthalpy is positive for both composition cases, the FCC phase stability over the BCC phase was mainly driven by enthalpy.

## Thermal stability of HEA NPs

To investigate the thermal stability of the HEA NPs produced, the samples were annealed at 1173 K (900 °C) for 72 h using a tube furnace with a continuous flow of argon. Their XRD patterns are shown in Supplementary Fig. 12 (Supplementary Note 1). After the annealing, the main phase (single FCC structure) of the HEA NPs still remained unchanged without peak splitting or new peak appearances, suggesting that dealloying or segregation of the constituent elements is minimal in both samples. However, a few minor phases with crystalline structures newly appeared. Based on the peak identification, it seems that those are most probably various oxide phases such as $Cr_2O_3$. Other minor phase is also observable with HEA-$H_2$ sample which might be from formation of $Co_7Mo_6$ or $Fe_7Mo_6$ intermetallic phase while this phase is absent in HEA-He sample suggesting different stabilities of the HEA NPs. This is attributable to the higher crystallinity of HEA-He sample. Overall, the HEA NPs produced by the plasma process are thermally stable even in a high-temperature environment of above 1000 K for an extended period of time.

## Growth mechanism of HEA NPs in a thermal plasma

To understand the in-situ plasma jet alloying mechanism, we have performed optical emission spectroscopy (OES), homogenous nucleation temperature calculations, and thermofluid simulations. Figure 4a shows an emission spectrum measured at Z = 0.23 m from the top of the plasma torch for the HEA-$H_2$ case. Atomic emission lines from the five elements are evident[21], indicating that the plasma jet temperature is high enough to vaporize the metal powders injected. Our numerical simulation predicts a temperature of above 8000 K at the plasma core (Fig. 4c). We note that the thermofluid simulations were performed without considering the injection of metals and thus the actual plasma temperature is expected to be slightly lower than the predicted one due to the powder loading effect.

A spectrum measured at Z = 0.49 m is presented in Fig. 4b. In this region, the plasma jet undergoes a rapid cooling by the jet expansion and the turbulence, caused by an abrupt diameter change at the entrance of the reactor (e.g., $D_r > 3D_t$, where $D_r$ is the reactor diameter and $D_t$ is the torch diameter). In this study, a reactor geometry that promotes turbulence has been employed to improve the intermixing of vapors produced from different metals (Fig. 4d). In Fig. 4e–g, the ratios of turbulent to laminar viscosity ($\mu_t / \mu_l$) are calculated for the different reactor geometries, and showed that the turbulence effect is enhanced compared to the $D_r = D_t$ case. Although the emission intensity in this region has decreased due to the cooling, all the atomic emission lines from the five elements are still evident. At the same time, the continuum emission from the formation of nanoparticles is negligible, suggesting that the temperature is still high to trigger either homogenous nucleation or co-nucleation. The temperature predicted at the center is around 4200 K. A similar behavior was observed with the spectra measured with the helium gas (Supplementary Fig. 13).

As the temperature decreases further, the saturation pressure of each metal vapor decreases and becomes comparable to its partial pressure (i.e., supersaturated, Fig. 4h), which leads to homogenous nucleation of vapors. For the given feed rate of 1.5 g min$^{-1}$, the homogeneous nucleation temperature of each metal was calculated using the self-consistent classical theory (Supplementary Note 2)[22–24]. Figure 4i presents the calculated nucleation temperature of each metal with melting temperatures of their bulks. Owing to their low partial pressures, most of the nucleation temperatures are close to or lower

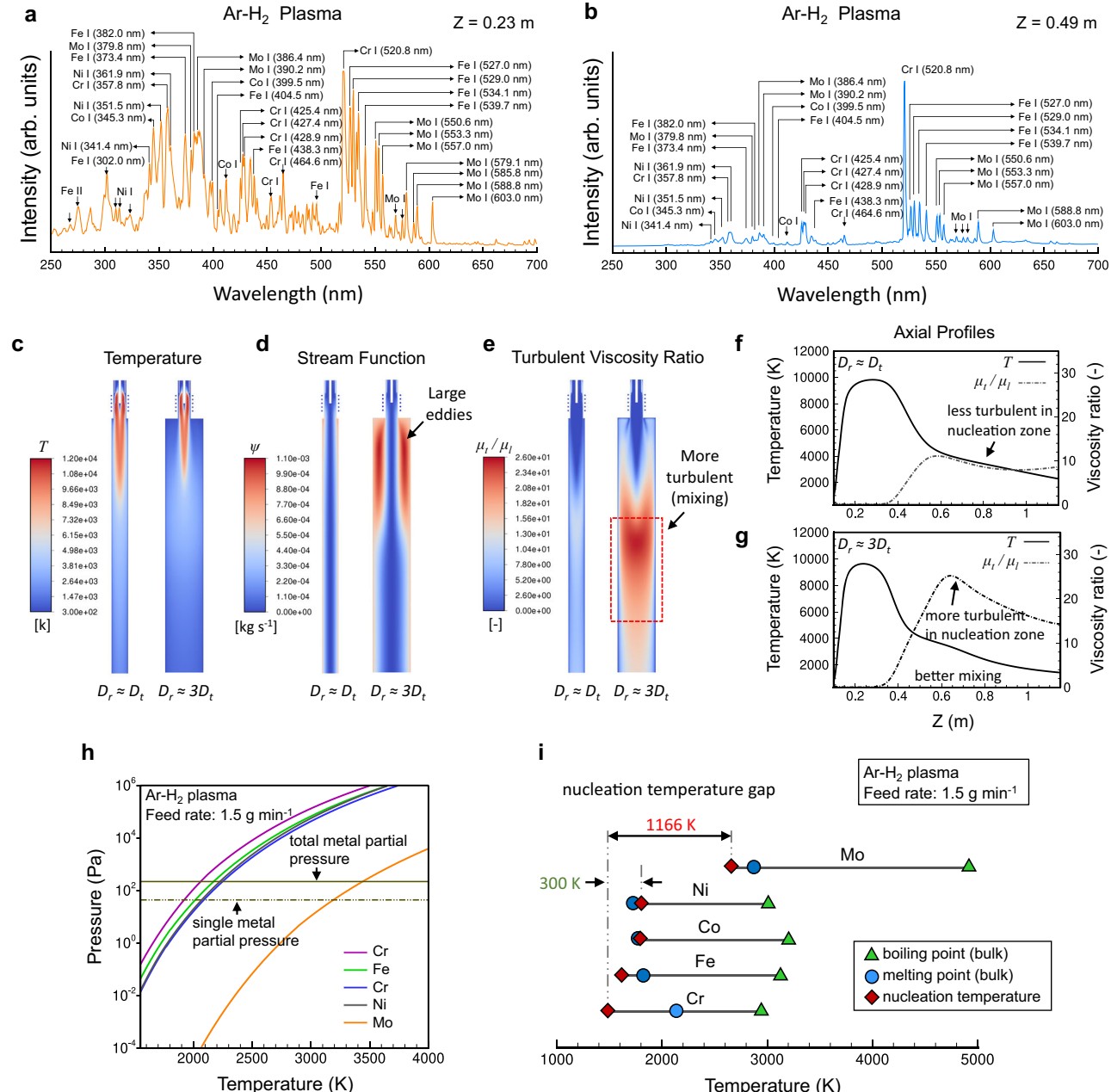

**Fig. 4 | Optical emission measurements, thermofluid simulations, and homogenous nucleation temperature calculations for the ICPJ process. a**, **b** Optical emission spectra measured at (**a**) $Z = 0.23$ m and (**b**) $Z = 0.49$ from the top of the plasma torch for the HEA-H$_2$ case. **c**–**g** Thermofluid simulation showing the effect of the reactor geometry ($D_r = D_t$ v.s. $D_r = 3D_t$ where $D_r$ is the reactor diameter and $D_t$ is

the torch diameter) on the turbulence intensity. **h**, **i** Calculated (**h**) saturation vapor pressures and (**i**) nucleation temperatures of each element in a vapor mix of Cr:Fe:Co:Ni:Mo = 1:1:1:1:1 produced at a feed rate of 1.5 g min$^{-1}$, showing the existence of a large nucleation temperature gap.

than the melting points of the bulk metals. Due to the low saturation vapor pressure of Mo (Fig. 4h)[20], Mo becomes supersaturated earlier than other metals and its nucleation temperature is estimated to be 2655 K and 2668 K for Ar-H$_2$ and Ar-He cases, respectively (see Supplementary Fig. 14 for the Ar-He case). The other metals nucleate between 1800 and 1500 K; thus, there exists a large nucleation temperature gap, over 1000 K, between Mo and the other metals. This gap may present a challenge in the formation of HEA particles via co-nucleation process. The melting temperature of the expected HEA NP was also estimated by using the mixing rule and considering the melting point depression effect because of the nanometer size (Supplementary Note 3 and Supplementary Table 6)[25,26]. For the given composition, it turned out to be 1865 K for case Ar-H$_2$ and 1919 K for

the Ar-He case, which is close to the nucleation temperatures of most metal elements except for Mo. Thus, in this ICPJ process, the growth mechanism of HEA NPs can be predominated either by VLS (vapor-liquid-solid) or VS (vapor-solid) transformation, depending on the size of the temperature zone of ~2000−2550 K. With a long residence time in this zone, it is proposed that Mo vapor forms nuclei first via homogenous nucleation around 2650 K, followed by co-condensation (i.e., heterogeneous condensation) of metal monomers (including Mo) to grow HEA NPs. However, it is unlikely that Mo nuclei continue to grow into pure Mo particles by homogenous condensation of Mo monomers due to its low partial pressure, which was decreased by mixing multiple elements at a near equimolar ratio (i.e., $P_i = P_{total}/N$). In addition, the particles formed reside in their liquid phase for enough

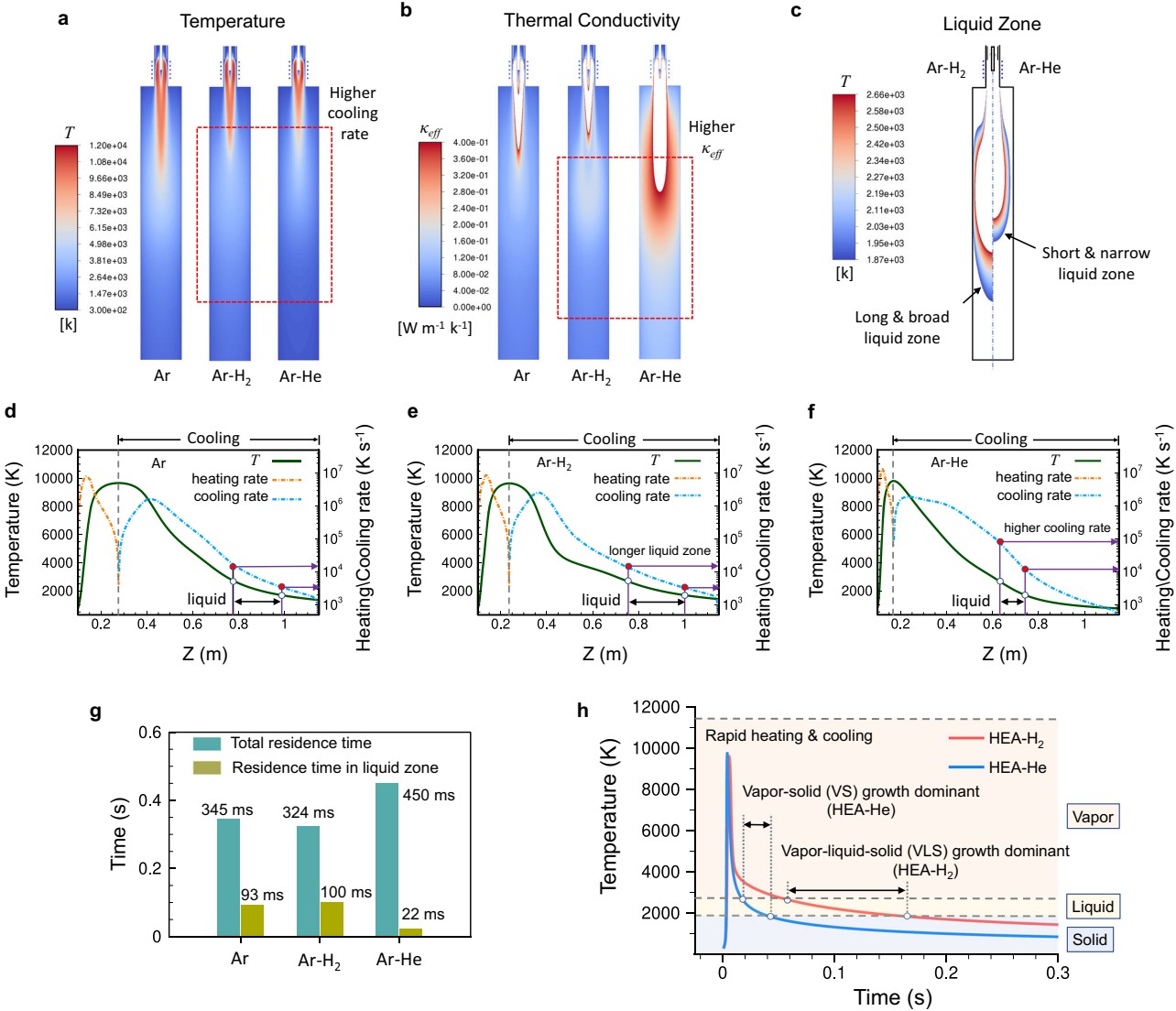

**Fig. 5 | Effects of the plasma gas on the HEA NP growth by the ICPJ strategy.**
**a** Temperature fields calculated for different plasma gases of Ar (100%), Ar-H$_2$ (H$_2$: 8.3%), and Ar-He (He: 77.4%), showing different cooling rates of the plasma jet. **b** Thermal conductivity ($\kappa_{eff}$) distributions calculated which show the effect of the plasma gas on the heat transfer rate. **c** Temperature zones where HEA NPs nucleated are expected to be in a liquid phase (i.e., liquid zone) for Ar-H$_2$ and Ar-He cases. **d–f** Axial temperature profiles with local heating and cooling rates calculated for different plasma gases. The arrows represent the estimated liquid zones. **g** Residence times of HEA NPs calculated for different plasma gases. **h** Growth mechanisms of HEA NPs under different plasma gases of Ar-H$_2$ and Ar-He.

time, and thus the different metal elements condensated can diffuse around in a particle creating homogenous mixing of the elements (i.e., alloying) driven by the high-entropy effect, until the temperature reaches the solidification limit of ~1890 K (i.e., VLS transformation). On the other hand, a small size of this temperature zone may result in a co-nucleation of five metals in the temperature range of 1800−1500 K, followed by co-condensation of their monomers. In this case, the direct vapor to solid transformation (i.e., VS transformation) would be the predominant NP growth mechanism as the co-nucleation temperature is lower than the melting temperature of HEA NPs estimated.

### Effect of plasma gases
The plasma jet temperature and the resulted thermofluid field can be controlled by carefully choosing the plasma gas composition because different gases have different heat transfer capabilities. Figure 5a shows the temperature fields inside the reactor calculated for different plasma gases of Ar (100%), Ar-H$_2$ (H$_2$: 8.3%), and Ar-He (He: 77.4%) (See Supplementary Fig. 15 for the velocity and species fields). Adding hydrogen or helium in the plasma gas alters the thermofluid field

substantially; it causes rapid cooling of the gas and consequently the plasma jet becomes shorter. The rapid quenching of the plasma jet can be explained by the higher thermal conductivity of hydrogen or helium over argon (Supplementary Fig. 16). Such gases can enhance the heat exchange with the surroundings downstream, resulting in rapid cooling of the plasma jet. Figure 5b shows the contours of thermal conductivities calculated. More enhanced heat transfer with the helium case is attributed to its higher concentration (77.4%) in the plasma gas compared with the hydrogen case (8.3%). The hydrogen content can be increased up to 20%; however, excessive hydrogen may have adverse effects on NPs such as hydrogen-driven embrittlement[27].

Complete evaporation of feedstock is a prerequisite to the formation of a homogeneous multicomponent vapor, which is largely affected by the heat exchange rate between plasma and feedstock and the residence time. Pure argon plasma presents the largest area of the hot zone which is favourable for feedstock evaporation while the heat exchange between feedstock and Ar plasma is expected to be slow due to the low thermal conductivity of Ar, resulting in a low vaporization efficiency. The hydrogen plasma seems more promising to achieve a

good evaporation efficiency thanks to its relatively high thermal conductivity and large hot zone, which is in line with the composition analysis.

In Fig. 5c, the temperature contour above the Mo nucleation temperature and below the HEA NP solidification temperature were cut-off. In this zone, Mo nuclei start to form from the homogenous nucleation and continue to grow into HEA NPs via the co-condensation of metal monomers. The particles are in the molten phase and thus can be alloyed by diffusion process inside particles. The size of this zone is larger for the hydrogen plasma case due to its moderate cooling rate. Hydrogen is also found to extend this zone towards the reactor exit, maintaining the gas temperature above the solidification limit by releasing the recombination heat of H atoms. The axial temperature profiles for each case are plotted in Fig. 5d–f with the corresponding local heating and cooling rates. In all cases, overall high heating and cooling rates up to $10^6 \, K \, s^{-1}$ were observed, which is ideal for the formation of HEA NPs. The liquid droplet zone is also indicated by an arrow and it appears earlier and lasts shorter in the helium plasma because of the faster cooling. The residence times along the centreline were calculated and presented in Fig. 5g. As expected, the longest residence time (100 ms) in the liquid droplet zone was observed with the hydrogen case while the shortest time (22 ms) was observed with the helium case. This may imply that NPs were grown via different pathways for the two different plasma gases.

From the thermofluid simulation, it is probable that the VLS transformation would be the predominant growth mechanism in the hydrogen case, while particles would grow via the direct VS transformation mechanism in the helium case (Fig. 5h). This could explain some differences in the structural and morphological properties between HEA-H$_2$ and HEA-He observed from the XRD, SEM, and TEM analysis. For the HEA-H$_2$ case, NPs grow through the liquid droplet phase, followed by rapid quenching. This allows for more formation of an amorphous or glass state (i.e., kinetically frozen liquid)[12], which is evident from the XRD analysis; the peak of (111) plane is broader than that of the HEA-He sample. The distribution of elements in a NP also seems more homogeneous and diffusive which is evident from the comparison of elemental maps. In the VLS growth, NPs can grow further through interparticle coagulation until they solidify. This increases the particle size as well as broadens the size distribution, which is in line with the SEM observation. For the HEA-He case, it is suggested that NP growth mostly occurs in the solidification limit through the direct VS transformation. Therefore, only stable phases, such as segregate phases or random alloys can be formed with less amorphous state[12]. This would be the reason why the HEA-He sample exhibit a higher crystallinity or crystallite size from the XRD patterns. The slight segregation of element observed (see Supplementary Fig. 6) also supports the VS growth mechanism. In the VS growth, NPs cannot grow via interparticle coagulation, resulting in size reduction with a narrow size distribution, which is in line with the SEM analysis. However, the HEA NPs produced via the VS path (i.e., the helium plasma case) exhibit a higher Mo loss in the composition due to the low vaporization efficiency of feedstock. Although the helium plasma provides a higher maximum temperature (9740 K) and gas conductivity compared to those of the hydrogen plasma, it cools down rapidly through the enhanced heat exchange with the surroundings. To achieve the VS growth without the large Mo losses, a lower feed rate of feedstock or nano-sized Mo powders can be employed to ensure a better evaporation Mo without altering the thermofluidic field significantly. Our study demonstrates that the structural or morphological properties of HEA NPs can be tailored in this plasma process by changing plasma gases which controls the thermal history of NPs effectively. It should be noted that the thermal history of NPs can also be controlled by other processing parameters, such as reactor geometry. For instance, a reactor geometry with a rapid jet expansion promotes the quenching of the reaction stream resulting in size reduction with a narrow size distribution. The presence of thermal insulators or active heating units inside the reactor can increase the residence time of NPs in the liquid zone reducing element segregation.

## Synthesis of other HEA NPs

The main purpose of the development of the ICPJ process is to demonstrate the scalable synthesis of non-precious metal based HEA NPs for their cost-effective applications. Therefore, in our study, we considered HEA systems mainly consisting of earth-abundant metals (e.g., Cr, Mn, Fe, Co, Ni, Cu, Mo) rather than precious metals. Molybdenum and Cu were included specifically because Mo has different physical properties compared with other elements (e.g., low saturation pressure) while Cu is known to be hardly alloyed with other elemental metals due to its positive binary mixing enthalpies (Supplementary Table 7)[28]. We further demonstrated HEA NPs of CrMnFeCoNi, MnFeCoNiCu, and CrFeCoNiCu, and their structural and compositional properties were studied by XRD and HR-TEM analyses including EDX elemental mapping and EDX line-scanning (Fig. 6, Supplementary Figs. 17–26). All the XRD patterns of the HEA NPs produced exhibit a single FCC structure (Supplementary Figs. 17, 20, 23; Supplementary Tables 8–10), indicating successful alloying of the feedstock materials. Two minor peaks show up around 46.4° and 48.6° in the CrMnFeCoNi sample synthesized with hydrogen and this would be attributable to the formation of metal hydrides. Those peaks are absent in the sample produced with helium. Figure 6 shows EDX elemental maps of a single and multiple HEA NPs synthesized with different plasma gases. For CrMnFeCoNi (Fig. 6a–d) and MnFeCoNiCu HEA NPs (Fig. 6e–h), homogenous mixing of five elements in a single particle was observed without significant elemental segregation or phase separation regardless of the plasma gases employed. The elements in CrMnFeCoNi and MnFeCoNiCu HEA NPs also have near equimolar ratios (Supplementary Table 11): Cr (20.2%), Mn (20.4%), Fe (20.7%), Co (20.3%), and Ni (18.4%) for CrMnFeCoNi HEA-H$_2$; Mn (17.8%), Fe (22.4%), Co (21.6%), Ni (19.2%), and Cu (18.9%) for MnFeCoNiCu HEA-H$_2$. On the other hand, the EDX elemental maps of CrFeCoNiCu HEA NPs indicate Cu segregation at the edges of particles while other elements of Cr, Fe, Co, and Ni are homogeneously distributed (Fig. 6i–l). This is attributable to the high positive binary mixing enthalpies between Cu and other elements, especially Cr and Fe (Supplementary Table 7). Similar observations have been reported in other HEA synthesis methods (e.g., arc discharge)[11] and other HEA systems of AlCrFeCoCu[29] and CoCrFeNiCuAl[30].

To study the uniformity of compositions in the ICPJ process, we have performed EDX analysis on 10-20 particles with different sizes for each HEA NP system (Supplementary Figs. 18, 21, 24 and Supplementary Table 11). The composition of the HEA NPs synthesized by the ICPJ process varies up to 8.4%. To study the spatial uniformity of compositions inside a single particle, we also performed EDX line scanning of individual particles (Supplementary Figs. 19, 22, 25). For CrMnFeCoNi and MnFeCoNiCu HEA NPs, uniform distribution of five elements across the particle was observed without significant elemental segregation, confirming the spatial uniformity of compositions throughout the particle. On the other hand, the EDX line scans of CrFeCoNiCu HEA NPs indicate Cu segregation at the edges of the particles (Supplementary Fig. 25a). We also found that the Cu segregation seems reduced as the particle size decreases (Supplementary Fig. 26). We speculate that a large particle may take more time to cool down and provide more time for the Cu segregation. In this case, the size of particle may affect the uniformity of compositions; a smaller particle size with a narrow size distribution would be favorable to achieve a good compositional uniformity. A torch or reactor geometry that allows a better control of the particle size and size distribution could be considered for further improvement.

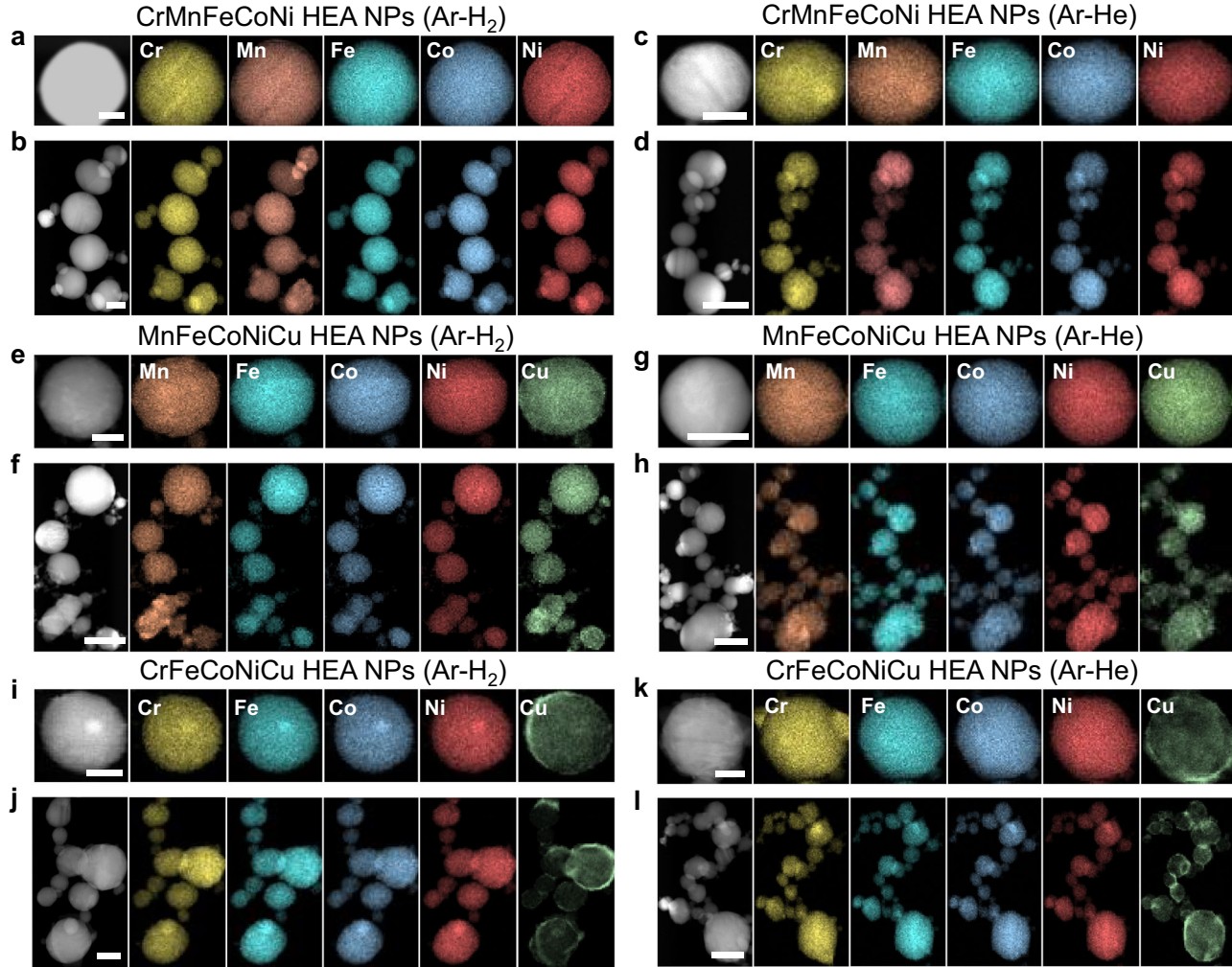

**Fig. 6 | HAADF-STEM images and EDX elemental maps of various HEA NPs synthesized by the ICPJ process. a, b** (**a**) a single and (**b**) multiple CrMnFeCoNi HEA NPs produced with hydrogen plasma. **c, d** (**c**) a single and (**d**) multiple CrMnFeCoNi HEA NPs produced with helium plasma. **e, f** (**e**) a single and (**f**) multiple MnFeCoNiCu HEA NPs produced with hydrogen plasma. **g, h** (**g**) a single and (**h**) multiple MnFeCoNiCu HEA NPs produced with helium plasma. **i, j** (**i**) a single and (**j**) multiple CrFeCoNiCu HEA NPs produced with hydrogen plasma. **k, l** (**k**) a single and (**l**) multiple CrFeCoNiCu HEA NPs produced with helium plasma. The elements in CrMnFeCoNi and MnFeCoNiCu HEA NPs have near equimolar ratios while the Cu content in CrFeCoNiCu HEA NPs (12-14%) is deviated from that of feedstock due to local segregation (Supplementary Figs. 18, 21, 24 and Supplementary Table 11). Scale bar, 50 nm (single NP) and 100 nm (multiple NPs).

## Optical absorption performance as a photothermal conversion material

Nanoparticles comprising multiple 3*d* transient-metal elements (e.g., Ti, V, Cr, Fe, Co, Ni, Cu) have been proven that their optical absorption can be broadened through the *d-d* interband transitions (IBTs), presenting a great potential as efficient photothermal conversion materials[31]. Such property of HEA NPs is attributable to the reinforcement of IBTs by fully filling energy regions around the Fermi level, upon introducing more 3*d* transient-metals in a nanoparticle. Very recently, this approach has also been extended to HEA NPs comprising beyond 3*d* transient-metals (e.g., Mo, Ta, W)[32]. Despite this new opportunity, the current synthesis method is intrinsically operated in a batch mode and not suitable for the preparation of HEA NPs for large-scale/area photothermal conversions[31,32]. Since our plasma jet process promises the commercial scale production of HEA NPs, the optical absorption performances of the as-produced HEA NPs have been investigated through a diffuse reflectance spectroscopic measurement in a wavelength region of 250 to 2500 nm (UV-Vis-NIR).

Figure 7 shows the absorptance spectra of the HEA NP samples along with that of the feedstock mixture. Both HEA NP samples exhibit an average absorptance greater than 96%, which is a significant improvement from that of the feedstock mixture. The strong absorption for the HEA NP samples is mainly contributable to *d-d* IBTs as reported previously[31,32]. Overall HEA-He sample shows better performance than HEA-H₂. Its higher crystalline structure or smaller size may be responsible for the enhanced performance; however, further understanding is needed to elucidate the mechanism for the improvement.

## Discussion

We develop an ultrafast (<100 ms), one-step method that enables the continuous synthesis of HEA NPs directly from a mixture of pure elemental metal powders using a high-temperature plasma jet (>5000 K). Highly crystalline HEA NPs (CrFeCoNiMo) were continuously produced at a high rate approaching 35 g h⁻¹ via in-flight alloying of elemental metal powders. This plasma process can be easily scaled up as the core technology is mature with high-power plasma torches up to ~MW levels available. Our study also indicates that the structural or morphological properties of HEA NPs can be tailored by changing the plasma gas which controls the thermal history of NPs. The yield demonstrated represents a seminal milestone towards the exploitation of HEA NPs in real-world applications.

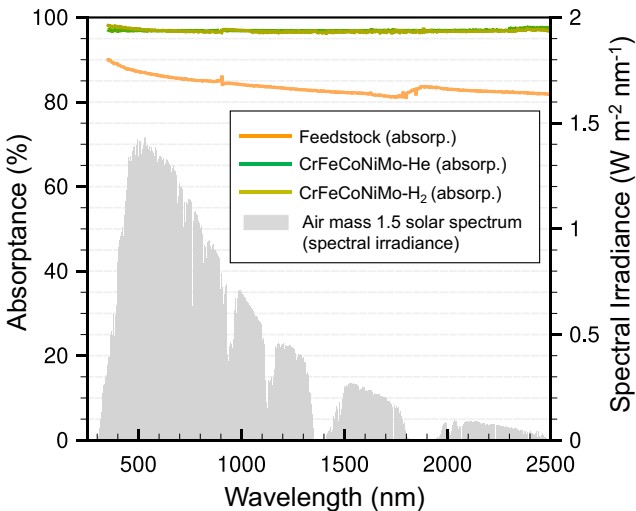

**Fig. 7 | Optical absorption performance of the HEA NPs.** Absorptance spectra of the HEA NPs (CrFeCoNiMo) prepared by the ICPJ strategy with different plasma gases. Both HEA NPs exhibit an excellent light absorption performance of >96%. The grey area presents the solar radiation spectrum (Air mass 1.5). The noisy signal above 2000 nm is due to water absorption bands, mostly in air. The discontinuity of the signal around 800 nm is caused by a detector change in the spectrophotometer.

The HEA NPs produced exhibit an excellent light absorption performance of >96%, and thus represent great potential in the cost-effective, large-area solar energy harvesting for thermo-photovoltaics, photocatalysis, and water desalination. In this regard, a direct solar energy absorption experiment with the HEA NPs produced in this work is of particular interest as future research.

## Methods

### Materials
Chromium (Cr, <10 μm, 99.2%), manganese (Mn, <10 μm, 99.6%), iron (Fe, 6–10 μm, 99.5%), cobalt (Co, 1.6 μm, 99.8%), nickel (Ni, 3–7 μm, 99.9%), copper (Cu, 0.5–1.5 μm, 99%), and molybdenum (Mo, 3–7 μm, 99.95%) powders were purchased from Alfa Aesar. The as-received elemental metal powders were mixed at an equal ratio (1:1:1:1:1), and the mixtures (i.e., Cr-Fe-Co-Ni-Mo, Cr-Mn-Fe-Co-Ni, Mn-Fe-Co-Ni-Cu, Cr-Fe-Co-Ni-Cu) were employed for the synthesis experiments without further treatment.

### Plasma synthesis system
HEA NPs were synthesized by using an RF thermal plasma technology. The plasma processing system was originally developed for the synthesis of nanotubes (e.g., carbon and boron nitride nanotubes)[33] and further modified for an effective synthesis of HEA NPs. The current synthesis system consists of five major parts: an induction plasma torch, a reaction chamber, a cyclone separator, a filtration chamber, and feedstock delivery (see Supplementary Fig. 1). For the plasma generation, a commercial RF induction plasma torch (Tekna PS-50, Tekna Systems, Inc.) composed of a five-turn coil and a ceramic tube with an internal diameter of 50 mm was employed. A 1-m long, double-walled stainless-steel chamber was employed as the reaction chamber. Its diameter was designed to be at least 3 times larger (e.g., 150 mm) than that of the plasma torch to enhance the mixing of metal vapors by the increased turbulence effect. The chamber walls were cooled by water to increase the cooling rate of the HEA NPs produced. To selectively remove unvaporized feedstock particles, a cyclone separator (inner diameter: 0.034 m; outer diameter: 0.097 m; length: 0.6 m) was employed at the bottom of

the reactor. The nano-size final products were collected from four porous metal filter units (surface area = 20 × 50 cm, 2.8 μm pore size) inside a filtration chamber connected to the end of the cyclone separator.

### HEA NP synthesis
In the synthesis experiment, the plasma power was fixed at 45 kW at an RF frequency of ~3 MHz (Lepel Co.) while two different plasma gas compositions were used to investigate the effect of the plasma gas: 5 slpm of carrier gas (Ar), 30 slpm of central gas (Ar), and 120/14 slpm of sheath gas (Ar/H$_2$); 5 slpm of carrier gas (Ar), 30 slpm of central gas (Ar), and 120 slpm of sheath gas (He). The feedstock was continuously fed by a vibrating powder feeder (PFR200 feeder, Tekna Systems, Inc.) and delivered to an injection probe located on the top of the plasma torch by Ar carrier gas. The feed rate of the powder mix was about 1.2–2.0 g min$^{-1}$. During the synthesis, the reactor pressure was kept constant at 66.7 kPa. The reaction products were collected from the cyclone separator and the filtration chamber in an open environment and characterized without further purification or treatment.

### Optical emission spectroscopy
To verify the formation of metal vapors from feedstock vaporization and investigate their spatial evolution, optical emission spectra were measured at Z = 0.23 m and Z = 0.49 m from the top of the plasma torch during the synthesis. A modular spectrometer (JAZ-EL200-XR1, Ocean Optics, with 1.7 nm FWHM resolution) was employed for the measurement over a wavelength range from 200 to 1025 nm. The emission light was collected through a quartz window and transported to the spectrometer by an optical fiber (QMMJ-55-UVVIS-200/240-2PCBL-0.25, OZ Optics Ltd., with a core size of 200 μm). It was confirmed that the emission from atomic oxygen (e.g., 777.4 nm) was not observable in our process, implying that oxygen contamination was not significant.

### Materials characterization
Morphology, structure and composition of the HEA NP samples produced under different conditions were analyzed by XRD, SEM and HR-TEM. SEM analysis was performed with a field emission scanning electron microscope (Hitachi S-4800). Samples were mounted on the sample holders with double-sided conductive glue tape and were imaged in a secondary electron mode. To avoid dislodgement of magnetic particles by magnetic objective lens, the working distance was chosen to be more than 9 mm. X-ray powder diffraction was performed using D8 DISCOVER (Bruker) with Cu $K_\alpha 1$ radiation ($\lambda = 1.5406$ Å, 8.0478 keV). The 2-theta range was set from 20 to 90 degree with a scan speed of 0.02 degree per second. TEM specimens were prepared by dispersing the solid powder in ethanol, and sonicating for 5 min. One drop of the solution was then placed onto a 400 mesh TEM copper/gold grid coated with an ultrathin (<3 nm) carbon film supported on lacey carbon (Ted Pella) and dried in air. A FEI Titan3 80–300 TEM operated at 300 keV, and equipped with a CEOS aberration corrector for the probe forming lens and a monochromated field-emission gun was used for the analytical TEM analysis. HAADF images were collected using a Fischione detector in scanning transmission electron microscopy (STEM) mode. The TEM instrument is also equipped with an energy-dispersive X-ray (EDX) spectrometer (EDAX Analyzer, DPP-II). To optimize the signal intensity, EDX spectra were acquired with the specimen tilted at 15 degrees.

### Light absorption measurement
The light absorptance spectra of the HEA NP samples were extracted from diffuse reflectance measurements. Diffuse reflectance

spectroscopic (DFS) measurements were done with a Perkin Elmer Lambda-900 spectrophotometer equipped with a 15 cm diameter integrating sphere (Labsphere). A home-made mortar and pestle system was used to crush powder samples into 1 cm diameter non-translucent pellets with a near Lambertian scattering behaviour, following a procedure similar to refs [34,35]. The reflectance was measured with the pellet still in the mortar, using a 1 cm diameter aperture accessory to expose only the surface of the pellet to the incident light beam. With the light beam cross-section being larger than the aperture, the reflectance of the aperture without samples ($I_O$) was subtracted from all measurements, and the samples measurements ($I_s$) were compared to a white standard ($I_{IOO}$, spectralon material, labsphere), given the diffuse reflectance as $R_d = (I_s\text{-}I_O)/(I_{IOO}\text{-}I_O)$. The absorptance $A$ was directly extracted using $A = 1\text{-}R_d + T_d = 1\text{-}R_d$, where $T_d$ is the transmittance, null is our case (the pellets were opaque). To reduce the spectral noise above 2000 nm and around 810 nm (detector change), we decrease the scan rate from 240 nm min$^{-1}$ to 6 nm min$^{-1}$ over these regions.

### Thermofluid simulations

To investigate the effects of reactor geometry and plasma gas composition on the thermofluid field (e.g., turbulence intensity, quenching rate, and residence time), computational fluid dynamics (CFD) simulations were performed inside the reactor using an in-house plasma code coupled with ANSYS-FLUENT[36]. The plasma generation model was based on the magneto-hydrodynamic (MHD) theory and consists of conservation equations for mass, momentum, and energy coupled with a set of Maxwell's equations. To investigate turbulence effects with reasonable computational costs, we adopted the Reynolds stress model which has been widely employed in the modeling of various nanomaterials synthesis by thermal plasmas[37]. Thermodynamic and transport properties of plasma gases (e.g., density, specific heat at constant pressure, viscosity, thermal conductivity, electrical conductivity, and radiation losses) were calculated under the local thermodynamic equilibrium (LTE) assumption[18]. Although powder injection and its vaporization are important phenomena to study, the vaporization of feedstock was not taken into account due to the lack of thermodynamic and transport data of some metal vapors for a wide range of temperature of 1000 – 10,000 K. More details on the governing equations, computational domain, and boundary conditions can be found in the Supplementary Note 4 (see Supplementary Fig. 27 and Supplementary Table 12).

### Density functional theory (DFT) simulations

The structure relaxation and total energy calculation were performed by using DFT implemented in the Vienna ab initio simulation package (VASP). The projector augmented wave (PAW) method and generalized gradient approximation (GGA) parameterized by Perdew, Burke and Ernzerhof (PBE) were used[38-40]. The basis set consists of plane waves with a cutoff energy of 520 eV. Monkhorst-Pack method with $1 \times 1 \times 2$ was used. Spin-polarized calculation were performed. For both BCC and FCC phases, 100-atom BCC and FCC supercell ($5 \times 5 \times 4$ unit cells) models were constructed. The random solid solution structures were generated by using the hybrid Cuckoo search (CS) code as implemented in ref [41]. Ten different random configurations for a given phase and composition were generated and examined. All the calculations were performed at 8 different lattice constants around the equilibrium, in which the unit cell shape was fixed and the atomic positions have been fully relaxed using conjugated-gradient algorithm.

## Data availability

All the supporting data are provided in the main text and Supplementary Information. The data sources that support the findings of this study are available from the corresponding author upon request.

## Code availability

The source code for the thermofluid simulation used in this study is managed by the National Research Council Canada (NRC-CNRC). It could be made available from the corresponding author for disclosure upon acceptance of NRC-CNRC's terms and conditions.

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

## Acknowledgements

This work was supported by the National Research Council Canada-University of Toronto Collaboration Centre in Green Energy Materials (CC-GEM) within the Materials for Clean Fuels (MCF) Challenge program at the National Research Council Canada (A1-018124, recipients K.S.K. and Y.Z.). The authors are grateful to R. Iannitto and G. Li for their assistant in the powder preparation and sharing their experiences on handling nano-sized metal powders.

## Author contributions

K.S.K. and H.S. conceived the idea and designed the present work. K.S.K., Z.T., D.R. and M.P. performed the HEA synthesis experiments. M.C., O.N. and L.G. carried out the detailed microscopic analyses. Z.T. and M.N. carried out the nucleation temperature calculation and XRD data interpretation. H.S. performed the DFT simulation on the phase stability. K.S.K. carried out the thermofluid simulation. D.P. conducted the light absorption measurement and extracted the absorptance spectra. T.L., O.N., C.C., L.C. and Y.Z. performed XRD characterization and electrochemical tests. All authors discussed the results together and commented on the manuscript.

## Competing interests

An international patent (PCT) application has been applied for via National Research Council Canada (No. PCT/CA2023/051556). The authors K.S.K, M.C, H.S, D.R, and M.P. are involved in the patent application. The remaining authors declare no competing interests.
