## [Peer Review File · Nature Communications]

Continuous Synthesis of High-entropy Alloy Nanoparticles by In-flight Alloying of Elemental MetalsEditorial Note: Parts of this Peer Review File have been redacted as indicated to maintain the confidentiality of unpublished data.

Reviewers' comments:

Reviewer #1 (Remarks to the Author):

The manuscript "Continuous Synthesis of High-entropy Alloy Nanoparticles by In-flight Alloying of Elemental Metals" reports the production of CrFeCoNiMo nanoparticles by an in-flight plasma jet technique. The employment of a high temperature (> 5000 K) plasma jet is proposed to achieve high heating and cooling rates, while the in-flight approach compared to the batch procedure allows the upscaling of the process achieving a productivity of 35 g/h. The synthesis of CrFeCoNiMo nanoparticles represents a challenge due to the segregation of its elemental constituents during the alloy formation. In order to address it, together with the fast cooling rates achieved by the high temperature plasma, the manuscript evaluates the effect of the plasma gases on the formation mechanism and elemental segregation. Finally, the optical properties of the CrFeCoNiMo nanoparticles are characterized, finding that the material exhibits a broad absorption spectrum from 200-2500 nm with an absorption > 96%.

The characterization of the nanoparticles by SEM, STEM, and XRD provide a complete overview of both the elemental composition of individual nanoparticles and the produced powder, observing the overall loss of Mo content while the Mo distribution in individual nanoparticles seems to be homogeneous. The DFT, and thermofluid simulations explain the stability of the produced FCC structures and provide a possible nanoparticle formation path as a function of the plasma gas employed, finding a vapour-liquid-solid or vapour-solid transition as a function of the residence time. This is related to the observed differences in size, and structure of the produced nanoparticles in H and He.

Overall, the manuscript reports a promising route towards industrial scale production of HEA nanoparticles. However, this general claim is only proved for a model material as CrFeCoNiMo, and the Mo loss observed indicates that the process cannot ensure that the targeted equiatomic CrFeCoNiMo nanoparticles are produced. Besides, some other points need to be clarified:

- To provide a clear view of the high productivity achieved by the in-flight plasma jet, the efficiency of the process needs to be mentioned in the abstract. In line 124 it is mentioned that 42 g are collected from the cyclone separator and 84 g from the filter unit. In the next sentence it is mentioned that 200 g of powder was fed to reach the 35 g/h productivity. Since the 42 g plus the 84 g are not matching the 200 g mentioned later, please clarify the exact amount of initial elemental powders added to the system, and the amount collected from it not converted into CrFeCoNiMo nanoparticles plus the amount converted into CrFeCoNiMo nanoparticles. A value for the process efficiency in terms of the initially added elemental powders is required.
- According to Fig. S5, the powders collected in the cyclone separator do not show exactly the same XRD spectra as the initial feedstock. Specifically, Cr powder seems to be in a lower amount or not present. Please clarify if the powder collected in the cyclone separator (unvaporized) can be reused in the process. Together with the efficiency of the process, the reusability of the elemental powders not converted into nanoparticles is a critical point to claim that the process can be transferred to industrial applications.
- As the authors mention, the Mo content in the cyclone separator is higher than other elements. This can explain the loss of Mo content in the produced nanoparticles, Fig. S10. How do the authors propose to account for these losses in order to produce the targeted nanoparticle composition?
- In Fig. 2c the fitting curves of the nanoparticle size distribution seem to indicate the presence of nanoparticles smaller than 10 nm. However, from Figs. 2b, Fig. 3a and b, and Figs. S3, S6, and S7 these nanoparticles are not observed. A log-normal fitting would be probably more accurate.
- In Fig. 3 a and b it is shown that the elemental distribution is homogeneous within the nanoparticles. Nevertheless, the targeted equiatomic composition is not achieved as mentioned in Figs. 3 g and h,

Table S3, and Fig. S10. To facilitate the reading, representative line scans of individual nanoparticles achieved in each gas (for example Fig. S7) should be added to Fig. 3 to make clear the amount of each element present.

- In line 302, the thermofluid simulations allow to conclude that the different formation paths (VLS and VS) in H₂ and He explain the smaller nanoparticle size, higher crystallinity, and segregation observed for the nanoparticles produced using He. This observation is interesting for the specific case of the CrFeCoNiMo nanoparticles produced in the gases proposed. To support the claim that the technique can be extended to other HEA nanoparticles production, a discussion of the options to control the nanoparticle size by adapting the temperature conditions, gas employed, and plasma reactor geometry should be provided.

- The residence time in liquid form in Fig. 5g is found to critically influence the nanoparticle formation mechanism. The authors find that higher residence times in liquid zone observed in H₂ reduce segregation. Strategies to be followed to increase this time, should be discussed.

- The VS formation path provides some features as higher crystallinity and smaller nanoparticle size. However, the nanoparticles produced using He (VS path) exhibit a high Mo loss. Please based on Fig. 5, comment the influence of the gas conductivity, plasma temperature profile, and maximum plasma temperature reached, on the elemental segregation, Mo vaporization, nanoparticle size, and crystallinity. Currently both discussions are not placed together. Would it be possible to achieve a VS formation path without the large Mo losses observed due to vaporization?

- The absorption spectrum shown in Fig. 6 is completely noise above 1900 nm. That data cannot support the statement of high absorption in that region, since most probably the intensity of the lamp employed in that region is quite low.

- Only the optical absorption spectrum is not enough to undoubtedly claim that the material will be good for large scale solar energy photothermal conversion applications. Thermal characterization of the material under irradiation are required to support that claim.

Reviewer #2 (Remarks to the Author):

The large scale HEA synthesis was demonstrated by using plasma jet. The characterization is good to proof that the fabrication method works. It is highly suspected that the temperature grading in the plasma jet can make the HEA particles not uniform in structure and composition. The relatively large size distribution in both plasma jets is one of the evidence and was discussed in the manuscript. However, it is still lack of the characterization and discussion of the uniformity of composition. I strongly encourage the team can have this analysis incorporated to the paper before the acceptance recommended. You might want to analyze a few tens of particles with different sizes to study the uniformity of compositions.

Reviewer #3 (Remarks to the Author):

In this study, new method of synthesis of High-entropy alloy NPs was developed using rapid heating with plasma jet. The author mentioned advantages in using plasma jet. But they synthesized only one component of CrFeCoNiMo alloy. Preparation condition is also limited. The evaluation of stability of HEA NPs is not enough. These small limited data are not enough to prove the advantages of this processes. Because of lack of data, this paper cannot be accepted to Nature Comm.

Response to reviews of NCOMMS-22-53178

**Editorial Office

We thank the reviewers for careful readings of our manuscript and valuable comments which improve the clarity of the manuscript. We also appreciate their favorable comments on the importance of this work towards further advance of scalable synthesis of High-entropy Alloy Nanoparticles (HEA NPs). We reproduced reviewer's comments below in *italic* font. Our reply is given below each point, and changes to the manuscript are indicated in **bold font**.

**Reviewer #1

We thank this reviewer for careful reading of our manuscript and the favorable comment that our work could provide a promising route towards industrial-scale production of HEA nanoparticles. We also appreciate this reviewer for his/her valuable comments on the control of HEA growth mechanisms, which helped us to improve the clarity of the manuscript. The main criticism from this Reviewer is the lack of data on other HEA systems, the work being limited to only one system. To fully incorporate this reviewer's feedback, in the revised manuscript, we included CrMnFeCoNi, MnFeCoNiCu, CrFeCoNiCu, and CrFeCoNiMo HEA NPs synthesized with different plasma gases of hydrogen and helium (in total of 8 samples). Structural and compositional properties of the newly synthesized HEA NPs have also been thoroughly studied by XRD and HR-TEM analysis, including EDX elemental mapping and EDX line-scanning (please see our Responses to Reviewer 2 and 3).

Below are our responses to this Reviewer's specific points.

Comment 1. *To provide a clear view of the high productivity achieved by the in-flight plasma jet, the efficiency of the process needs to be mentioned in the abstract. In line 124 it is mentioned that 42 g are collected from the cyclone separator and 84 g from the filter unit. In the next sentence it is mentioned that 200 g of powder was fed to reach the 35 g/h productivity. Since the 42 g plus the 84 g are not matching the 200 g mentioned later, please clarify the exact amount of initial elemental powders added to the system, and the amount collected from it not converted into CrFeCoNiMo nanoparticles plus the amount converted into CrFeCoNiMo nanoparticles. A value for the process efficiency in terms of the initially added elemental powders is required.*

We thank the reviewer for this valuable comment on the process efficiency. Below is a summary of the mass balance and conversion efficiency in our plasma process.

Amount of powder fed	Running time	Powder collected			Productivity	Conversion efficiency
		reactor	cyclone	filter		
200 g	150 min	74 g	42 g	84 g	33.6 g/h	42 %

In our current setup, a fair amount of powders (74 g) deposit on the reactor wall at downstream. The reactor wall is cooled by cold water which usually promotes diffusion of small powders towards reactor walls through thermophoretic forces. This may understate the conversion efficiency of our plasma process by lowering the powder collection efficiency.

Because of this issue, in the industrial-scale plasma process, a reactor with porous wall is usually employed. In such configuration, an inert gas is continuously flown through the porous wall and thus the powder deposition on the reactor walls can be minimized. We expect that such strategy can increase the conversion efficiency of our plasma process further by improving the powder collection efficiency.

In the revised manuscript, we mentioned the conversion efficiency in the abstract and provided the details on the mass balance in our plasma process in the first paragraph of page 6 as well as in the Supplementary Information (Supplementary Table 1).

***Comment 2.** According to Fig. S5, the powders collected in the cyclone separator do not show exactly the same XRD spectra as the initial feedstock. Specifically, Cr powder seems to be in a lower amount or not present. Please clarify if the powder collected in the cyclone separator (unvaporized) can be reused in the process. Together with the efficiency of the process, the reusability of the elemental powders not converted into nanoparticles is a critical point to claim that the process can be transferred to industrial applications.*

We thank the reviewer for this critical comment on reusability of the powders collected in the cyclone separator. As this reviewer indicated, the XRD patterns of those powders are different from that of feedstock because of 1) the different vapor pressures (or vaporization efficiencies) among different elements and 2) the partial alloying effect. For the latter case, we could recycle those powders in our plasma process and complete the alloying process; however, the former case may impose a challenge in reusing those powders as the ratio among different elements has changed. In this case, the vaporization efficiency should be improved through optimization of the processing parameters (e.g., feed rate, plasma power, powder injection geometry and so on) so that the amount of powders collected in the cyclone can be minimized.

Since we agree with this reviewer's comment that this would be a critical point to claim that our process can be transferred to industrial applications, we will continue to pursue this opportunity in our future work.

In the first paragraph of page 7, we mentioned the limitation in reusing the powders collected from the cyclone separator and also indicated that the vaporization efficiency should be improved through the process optimization.

Comment 3. As the authors mention, the Mo content in the cyclone separator is higher than other elements. This can explain the loss of Mo content in the produced nanoparticles, Fig. S10. How do the authors propose to account for these losses in order to produce the targeted nanoparticle composition?

A similar challenge has been reported in other HEA NP synthesis methods (e.g., arc discharge, microwave heating) where different metals with a large vapor pressure gap were employed [7,11]. A higher plasma power would be one of the viable solutions to address this challenge ensuring a better evaporation of Mo in feedstock.

On the other hand, the temperature–vapor pressure–mass relationship can be used to achieve the targeted composition as discussed in ref [7], e.g., increase the concentration of Mo in feedstock to compensate its low vapor pressure. In this regard, we have performed several synthesis experiments with different Mo contents in feedstock (e.g., 20 at%, 30 at%, and 40 at%) and provided the supporting data in the Supplementary Information only for review purpose (Supplementary Figs. 28 and 29). The EDX composition analysis show that the Mo composition in the products increases with the Mo content in feedstock; however, the XRD patterns indicate that those powders were not fully alloyed and also contaminated unvaporized feedstock. It seems that other processing conditions also need to be adjusted accordingly to accommodate the high Mo content in feedstock, e.g., reduce the feed rate and/or increase the plasma power. Since our plasma facility is currently under renovation, we couldn't investigate further at this moment. We will continue to work on this challenge as soon as our plasma facility is accessible.

In the first paragraph of page 8, we mentioned that, to achieved the targeted composition, we could increase the plasma power to ensure the complete vaporization of Mo or increase the Mo content in feedstock to compensate its low evaporation pressure.

Comment 4. In Fig. 2c the fitting curves of the nanoparticle size distribution seem to indicate the presence of nanoparticles smaller than 10 nm. However, from Figs. 2b, Fig. 3a and b, and Figs. S3, S6, and S7 these nanoparticles are not observed. A log-normal fitting would be probably more accurate.

We thank the reviewer for this valuable comment on the curve fitting of the nanoparticle size distribution. We have revised the fitting curves using a log-normal fitting method.

In Fig. 2, we provided fitting curves (Fig. 2c) revised using a log-normal fitting method.

Comment 5. In Fig. 3 a and b it is shown that the elemental distribution is homogeneous within the nanoparticles. Nevertheless, the targeted equiatomic composition is not achieved as mentioned in Figs. 3 g and h, Table S3, and Fig. S10. To facilitate the reading, representative line scans of individual nanoparticles achieved in each gas (for example Fig. S7) should be added to Fig. 3 to make clear the amount of each element present.

We thank the reviewer for this valuable suggestion. To facilitate the reading, we have included representative EDX line scans of individual nanoparticles for each HEA NP, which also show a good spatial uniformity in the composition.

In Fig. 3, we provided representative EDX line scans of individual nanoparticles produced with different plasma gases (Figs. 3i and j).

Comment 6. In line 302, the thermofluid simulations allow to conclude that the different formation paths (VLS and VS) in H₂ and He explain the smaller nanoparticle size, higher crystallinity, and segregation observed for the nanoparticles produced using He. This observation is interesting for the specific case of the CrFeCoNiMo nanoparticles produced in the gases proposed. To support the claim that the technique can be extended to other HEA nanoparticles production, a discussion of the options to control the nanoparticle size by adapting the temperature conditions, gas employed, and plasma reactor geometry should be provided.

We thank the reviewer for his/her positive comment on our findings. In this work, we demonstrated that the growth of HEA NPs can be controlled by employing different plasma gases in our plasma process. As commented by this reviewer, other parameters such as the reactor geometry can also affect the nanoparticle growth by altering the thermal history of NPs effectively. In this regard, we have been working on HEA NP synthesis with a different reactor geometry. [REDACTED]

[REDACTED]

[REDACTED]

In the first paragraph of page 14, we briefly mentioned that the reactor geometry can also be altered to control the growth of HEA NPs.

Comment 7. The residence time in liquid form in Fig. 5g is found to critically influence the nanoparticle formation mechanism. The authors find that higher residence times in liquid zone

observed in H2 reduce segregation. Strategies to be followed to increase this time, should be discussed.

We thank the reviewer for this valuable suggestion. In this study, we proposed that hydrogen naturally extends this liquid zone thanks to the recombination heat released from H atoms. We could also extend this zone further by employing a reactor with thermal insulator or active heating units. A reactor geometry with a large diameter can also be considered because it decreases the gas velocity increasing the residence time of NPs. However, it should be noted that an overly extended residence time may prolong the particle growth through coagulation so may increase the particle size and also broaden the particle distribution. An optimum range should exist. We agree that this would be a critical point that we need to investigate in our future study.

In the first paragraph of page 14, we discussed how the residence time in the liquid zone can be extended.

***Comment 8.** The VS formation path provides some features as higher crystallinity and smaller nanoparticle size. However, the nanoparticles produced using He (VS path) exhibit a high Mo loss. Please based on Fig. 5, comment the influence of the gas conductivity, plasma temperature profile, and maximum plasma temperature reached, on the elemental segregation, Mo vaporization, nanoparticle size, and crystallinity. Currently both discussions are not placed together. Would it be possible to achieve a VS formation path without the large Mo losses observed due to vaporization?*

We thank the reviewer for this valuable comment to improve the clarity of our findings. From the third paragraph of page 13, we have discussed the influence of the gas conductivity, plasma temperature profile, and maximum plasma temperature reached, on the elemental segregation, nanoparticle size, and crystallinity but not on the Mo vaporization. As suggested by this reviewer, the discussion is now followed by a discussion on the Mo losses to facilitate reading.

We also thank the reviewer for the interesting question on how to minimize the Mo loss in the VS formation path. To achieve the VS growth without the large Mo losses, we may employ a lower feed rate of feedstock or nano-sized Mo powder to ensure a better evaporation Mo without altering the thermofluidic field significantly. We believe this would be an interesting future study.

In the first paragraph of page 14, the both discussions are placed together to facilitate the reading and we also suggested that a lower feed rate of feedstock or nano-sized Mo powder can be employed to achieve the VS growth without the large Mo losses.

***Comment 9.** The absorption spectrum shown in Fig. 6 is completely noise above 1900 nm. That data cannot support the statement of high absorption in that region, since most probably the intensity of the lamp employed in that region is quite low.*

We thank the reviewer for this comment. We repeated the measurements and tried to reduce the noise as much as possible by purging nitrogen inside the spectrophotometer and using a much slower scan rate (6 nm/min) for the spectral regions where noise was observed. Still some noise remains and is due to a combination of sphere and detectors response and humidity.

In Fig. 6 (now Fig. 7), new data obtained with less noise was provided.

Comment 10. Only the optical absorption spectrum is not enough to undoubtedly claim that the material will be good for large scale solar energy photothermal conversion applications. Thermal characterization of the material under irradiation are required to support that claim.

We thank the reviewer for this critical comment and share his/her point of view. Although the optical absorption measurement can indicate potential of the HEA NPs in the large-scale solar energy harvest, a direct solar energy absorption performance may be required to fully support the claim. In the literature, a water evaporation rate was usually measured to evaluate photothermal conversion performance of materials using a simulated sun light (e.g., a Xe light with an AM 1.5 filter) and a full-spectrum optical power meter. It would be beneficial to include thermal characterization data but such set-up is not currently available in our lab. We proposed a direct solar energy absorption experiment as future research.

In Conclusion, we replaced the sentence of “will make an impact” with “represent great potential” and also suggested a direct solar energy absorption experiment as future research.

**Reviewer #2

Comment. The large scale HEA synthesis was demonstrated by using plasma jet. The characterization is good to proof that the fabrication method works. It is highly suspected that the temperature grading in the plasma jet can make the HEA particles not uniform in structure and composition. The relatively large size distribution in both plasma jets is one of the evidence and was discussed in the manuscript. However, it is still lack of the characterization and discussion of the uniformity of composition. I strongly encourage the team can have this analysis incorporated to the paper before the acceptance recommended. You might want to analyze a few tens of particles with different sizes to study the uniformity of compositions.

We thank the reviewer for careful reading of our manuscript and the favorable comment on our

work. We also appreciate this reviewer for his/her critical comment on our uniformity analysis. To thoroughly study the uniformity of composition in our plasma process, as suggested by this reviewer, we have performed EDX analysis on 10~20 particles with different sizes, for 4 different HEA systems (i.e., CrFeCoNiMo, CrMnFeCoNi, MnFeCoNiCu, CrFeCoNiCu) synthesized with different plasma gases of hydrogen and helium (in total of 8 samples, Supplementary Fig. 10, 18, 21, 24 and Table 11).

As summarized in the table below (Supplementary Table 11), the composition of the HEA NPs synthesized by our plasma process varies up to 8.4 % (Mo contents in CrFeCoNiMo HEA NPs). As indicated by this Reviewer, the variation is attributable to the relatively large particle size distribution rather than the temperature gradient in the plasma jet. The process may need to be optimized in this regard.

Ar-H ₂ plasma	Cr (%)	Mn (%)	Fe (%)	Co (%)	Ni (%)	Cu (%)	Mo (%)
CrMnFeCoNi	20.2 ± 3.6	20.4 ± 4.7	20.7 ± 1.8	20.3 ± 1.5	18.4 ± 2.1		
MnFeCoNiCu		17.8 ± 4.7	22.4 ± 3.4	21.6 ± 4.1	19.2 ± 3.6	18.9 ± 6.9	
CrFeCoNiCu	20.5 ± 2.8		22.1 ± 1.7	23.3 ± 3.1	22.1 ± 2.8	11.7 ± 5.6	
CrFeCoNiMo	19.2 ± 4.2		21.5 ± 3.2	22.3 ± 1.8	22.1 ± 3.2		14.7 ± 8.4

Ar-He plasma	Cr (%)	Mn (%)	Fe (%)	Co (%)	Ni (%)	Cu (%)	Mo (%)
CrMnFeCoNi	18.9 ± 2.6	19.9 ± 6.6	21.6 ± 2.2	22.0 ± 2.9	17.3 ± 2.8		
MnFeCoNiCu		15.9 ± 4.8	23.5 ± 1.9	25.1 ± 4.2	19.5 ± 3.0	16.0 ± 3.1	
CrFeCoNiCu	20.5 ± 3.4		22.1 ± 2.4	23.4 ± 2.8	20.4 ± 2.5	14.0 ± 6.7	
CrFeCoNiMo	23.9 ± 2.4		23.3 ± 1.6	25.6 ± 1.9	22.0 ± 0.9		5.2 ± 1.0

To study the spatial uniformity of compositions inside particles, we have also performed EDX elemental mapping for a single and multiple particles with different sizes (please see our Response to Reviewer Comment 3.1) as well as EDX line scanning of individual particles (please see below).

For CrMnFeCoNi, MnFeCoNiCu, and CrFeCoNiMo HEA NPs, homogenous mixing of five elements in a single particle was observed without significant elemental segregation or phase

separation regardless of the plasma gases employed, confirming the good spatial uniformity of compositions in our plasma process. The elements in CrMnFeCoNi and MnFeCoNiCu HEA NPs also have near equimolar ratios. On the other hand, the EDX elemental maps and line scans of CrFeCoNiCu HEA NPs indicate Cu segregation at the edges of particles while other elements of Cr, Fe, Co, and Ni are homogeneously distributed. This is attributable to the high positive binary mixing enthalpies between Cu and other elements, especially Cr and Fe [1]. Similar observations have been reported in other HEA synthesis method (e.g., arc discharge [2]) and other HEA systems of AlCrFeCoCu [3] and CoCrFeNiCuAl [4]. We also found that the Cu segregation seems reduced as the particle size decreases (Supplementary Fig. S26). We speculate that a large particle may take more time to cool down and provide more time for the Cu segregation. In this case, the particle size may affect the uniformity of compositions; smaller particles with a narrow size distribution would be favorable to achieve a good composition uniformity.

[REDACTED]

In the revised manuscript, we have provided more data on the statistical analysis of composition (Supplementary Fig. 10, 18, 21, 24 and Table 11) along with EDX elemental maps (Fig. 6) and EDX line scans (Fig. 3, Supplementary Fig. 19, 22, and 25), for 4 different HEA systems (i.e., CrFeCoNiMo, CrMnFeCoNi, MnFeCoNiCu, CrFeCoNiCu) synthesized with different plasma gases of hydrogen and helium (in total of 8 samples). In the first paragraph of page 8, we mentioned that the variation is attributable to the relatively large particle size distribution rather than the temperature gradient. In the second paragraph of page 15, we also discussed potential effect of the particle size distribution on the uniformity of compositions in our plasma process.

**Reviewer #3

***Comment.** In this study, new method of synthesis of High-entropy alloy NPs was developed using rapid heating with plasma jet. The author mentioned advantages in using plasma jet. But they synthesized only one component of CrFeCoNiMo alloy. Preparation condition is also limited. The evaluation of stability of HEA NPs is not enough. These small limited data are not enough to prove the advantages of this processes. Because of lack of data, this paper cannot be accepted to Nature Comm.*

(1) More HEA systems:

We thank the reviewer for careful readings of our manuscript and his/her critical comment that our study is limited to only one system of CrFeCoNiMo alloy. To fully incorporate this reviewer's feedback, we have demonstrated more HEA systems with targeted compositions

including CrMnFeCoNi, MnFeCoNiCu, and CrFeCoNiCu HEA NPs with different plasma gases of hydrogen and helium. The main purpose of the development of this plasma process is to demonstrate the scalable synthesis of non-precious metal based HEA NPs for their cost-effective applications. Therefore, in our work, we have considered HEA systems mainly consisting of earth-abundant metals (e.g., Cr, Mn, Fe, Co, Ni, Cu, Mo) rather than precious metals. Molybdenum and Cu were included specifically because Mo has different physical properties compared with other elements including large atomic size, high melting temperatures, and low saturation pressure while Cu is known to be hardly alloyed with other metals due to its positive binary mixing enthalpies [1]. In the revised manuscript, structural and compositional properties of the newly synthesized HEA NPs have been thoroughly studied by XRD and HR-TEM analysis, including EDX elemental mapping and EDX line-scanning.

All the XRD patterns of the HEA NPs produced exhibit a single FCC structure (Supplementary Figs. 17, 20, and 23; Supplementary Tables 8-10), indicating successful alloying of the feedstock materials. Two minor peaks show up around 46.4° and 48.6° in the CrMnFeCoNi sample synthesized with hydrogen and this is attributable to the formation of metal hydrides. Those peaks are absent in the sample produced with helium.

Mixing enthalpy of binary elementary combinations

	Co	Cr	Cu	Fe	Mn	Mo	Ni
Co	-	-4	6	-1	-5	-5	0
Cr	-4	-	12	-1	-2	0	-7
Cu	6	12	-	13	4	19	4
Fe	-1	-1	13	-	0	-2	-2
Mn	-5	-2	4	0	-	5	-8
Mo	-5	0	19	-2	5	-	-7
Ni	0	-7	4	-2	-8	-7	-

unit: kJ/mol

From the EDX mapping, homogenous mixing of five elements in a single particle was observed for CrMnFeCoNi and MnFeCoNiCu HEA NPs, without significant elemental segregation or phase separation regardless of the plasma gases employed. The elements in these HEA NPs also have near equimolar ratios (Supplementary Table 11):

Ar-H ₂ plasma	Cr (%)	Mn (%)	Fe (%)	Co (%)	Ni (%)	Cu (%)	Mo (%)
CrMnFeCoNi	20.2	20.4	20.7	20.3	18.4	-	-
MnFeCoNiCu	-	17.8	22.4	21.6	19.2	18.9	-
CrFeCoNiCu	20.5	-	22.1	23.3	22.1	11.7	-

On the other hand, the EDX elemental maps and line scans of CrFeCoNiCu HEA NPs indicate Cu segregation at the edges of particles while other elements of Cr, Fe, Co, and Ni are homogeneously distributed. This is attributable to the high positive binary mixing enthalpies between Cu and other elements, especially Cr and Fe [1]. Similar observations have been reported in other HEA synthesis method (e.g., arc discharge [2]) and other HEA systems of AlCrFeCoCu [3] and CoCrFeNiCuAl [4]. The Cu content in these samples is also deviated from that of the feedstock due to the local segregation. We also found that the Cu segregation seems reduced as the particle size decreases (please see below and Supplementary Fig. S26). We speculate that a large particle may take more time to cool down and provide more time for the Cu

segregation. In this case, the particle size may affect the uniformity of compositions; smaller particles with a narrow size distribution would be favorable to achieve a good composition uniformity.

[REDACTED]

In the revised manuscript, we provided more examples of HEA NPs (e.g., CrMnFeCoNi, CrFeCoNiCu, and MnFeCoNiCu) synthesized by this plasma method along with their characterization including XRD (Supplementary Figs. 17, 20, and 23; Supplementary Tables 8-10), EDX elemental maps (Fig. 6), statistical analysis on their compositions (Supplementary Figs. 18, 21, 24 and Table 11), and EDX line scans (Supplementary Figs. 19, 22, 25, and 26). In the second paragraph of page 14, we discussed the main results.

(2) Stability of HEA NPs:

The phase stability of CrFeCoNiMo HEA NPs produced with different plasma gases (i.e., HEA-H₂ and HEA-He) was extensively discussed in our original manuscript using DFT simulations. Based on the calculations, we also estimated the temperatures that satisfy the solid-solution criteria (i.e., the negative Gibbs free energy) for each case. To study the thermal stability of the HEA NPs produced, we annealed CrFeCoNiMo HEA NPs at 1,173 K (900 °C) for 72 hours

using a tube furnace with a continuous flow of argon. Their XRD patterns are shown in Supplementary Fig. 12. After the annealing, the main phase (single FCC structure) of the HEA NPs still remained unchanged without peak splitting or new peak appearances, suggesting that dealloying or segregation of the constituent elements is minimal in both cases. However, few minor phases with crystalline structures newly appeared. Based on the peak identification, it seems that those are most probably various oxide phases such as Cr_2O_3 . Other minor phase is also observable with HEA- H_2 which might be from formation of Co_7Mo_6 or Fe_7Mo_6 intermetallic phase while this phase is absent in HEA-He suggesting different stabilities of the HEA NPs. This is attributable to the higher crystallinity of the HEA-He sample. Overall, the HEA NPs produced by this plasma process are thermally stable even in a high-temperature environment of above 1,000 K for an extended period of time.

We provided XRD data of CrFeCoNiMo HEA NPs which were annealed at 1,173 K (900 °C) for 72 hours under an argon atmosphere (Supplementary Fig.12) and discussed their thermal stability in the second paragraph of page 9.

References

1. Takeuchi, A. & Inoue, A. Calculations of mixing enthalpy and mismatch entropy for ternary amorphous alloys. *Materials transactions, JIM*, **41**, 372-1378 (2000).
2. Mao, A. et al. Plasma arc discharge synthesis of multicomponent Co-Cr-Cu-Fe-Ni nanoparticles. *J. Alloys Compd.* **775**, 1177-1183 (2019).
3. Zhang, K. B. et al. Annealing on the structure and properties evolution of the CoCrFeNiCuAl high-entropy alloy. *J. Alloys Compd.* **502**, 295-299 (2010).
4. Tsai, M.H. et al. Morphology, structure and composition of precipitates in $\text{Al}_{0.3}\text{CoCrCu}_{0.5}\text{FeNi}$ high-entropy alloy. *Intermetallics* **32**, 329-336 (2013)

REVIEWERS' COMMENTS

Reviewer #1 (Remarks to the Author):

All of my comments have been appropriately addressed by the authors, leading me to recommend the publication of the manuscript.

Reviewer #2 (Remarks to the Author):

The team did extensive work to synthesize more samples with different compositions. Also STEM analysis was carried out to check the uniformity of the size and composition. The comments were addressed well. Acceptance for publication is recommended.

Reviewer #3 (Remarks to the Author):

Because moderate modifications were made, this paper can be accepted.